# Melatonin in *Brassicaceae*: Role in Postharvest and Interesting Phytochemicals

**DOI:** 10.3390/molecules27051523

**Published:** 2022-02-24

**Authors:** Josefa Hernández-Ruiz, Domingo Ruiz-Cano, Manuela Giraldo-Acosta, Antonio Cano, Marino B. Arnao

**Affiliations:** Department of Plant Biology (Plant Physiology), Faculty of Biology, University of Murcia, 30100 Murcia, Spain; jhruiz@um.es (J.H.-R.); domingo.ruiz1@um.es (D.R.-C.); manuela.giraldoa@um.es (M.G.-A.); aclario@um.es (A.C.)

**Keywords:** broccoli, melatonin, postharvest, glucosinolates, *Brassica*, fresh cut, vegetables

## Abstract

Brassicaceae plants are of great interest for human consumption due to their wide variety and nutritional qualities. Of the more than 4000 species that make up this family, about a hundred varieties of 6–8 genera are extensively cultivated. One of the most interesting aspects is its high content of glucosinolates, which are plant secondary metabolites with widely demonstrated anti-oncogenic properties that make them healthy. The most relevant Brassicaceae studies related to food and melatonin are examined in this paper. The role of melatonin as a beneficial agent in seedling grown mainly in cabbage and rapeseed and in the postharvest preservation of broccoli is especially analyzed. The beneficial effect of melatonin treatments on the organoleptic properties of these commonly consumed vegetables can be of great interest in the agri-food industry. Melatonin application extends the shelf life of fresh-cut broccoli while maintaining optimal visual and nutritional parameters. In addition, an integrated model indicating the role of melatonin on the organoleptic properties, the biosynthesis of glucosinolates and the regulatory action of these health-relevant compounds with anti-oncogenic activity is presented.

## 1. Introduction

Melatonin (*N*-acetyl-5-methoxytryptamine) is an indolic bioamine derived from tryptophan as a precursor in its biosynthesis. Since its discovery in 1958 in the pineal gland of cow [1,2], its role as an animal hormone in vertebrates has been intensively studied. Multiple studies have shown a relevant role of melatonin in various cellular and physiological actions as a regulator of wake–sleep cycles, body temperature, mood, sexual behavior, some endocrine rhythms and neuronal and immunological activity. The most recent studies emphasize the important role of this molecule in glucose metabolism and insulin action. Also noteworthy are the studies on melatonin’s anti-oncogenic role in many tumors as a sensitizer in chemical and radiological therapies. Moreover, its therapeutic efficacy in Parkinson’s and Alzheimer’s disease as well as in COVID-19 has also been studied [3,4,5,6,7,8,9,10,11,12].

In 1995, the view of melatonin as an hormonal molecule in mammals underwent a sudden change. Melatonin was identified undoubtedly in several plants and then it was considered as a universal biological molecule, since later it was also identified in invertebrates, fungi and bacteria [13,14,15,16,17]. In plants, numerous studies in the last two decades have been carried out to provide a significant body of research on the actions of melatonin on the physiology of plants [18]. Currently, we can consider that melatonin, called phytomelatonin in plants, performs multiple actions, modulating practically all the responses of plants [18,19]. Thus, phytomelatonin intervenes and/or improves processes such as germination, rooting, growth, flowering and fruiting [20,21,22,23,24,25,26]. The most obvious actions of phytomelatonin are evident in situations of stress [27]. Phytomelatonin improves the response and tolerance to both abiotic and biotic stressors. This function as a biostimulant seems to be related to its multiple regulatory actions on the primary and secondary metabolism of plant cells, possibly through its action on the homeostasis of the redox network and biological rhythms [28,29,30,31,32,33,34,35].

Phytomelatonin is considered a plant master regulator because it seems to regulate the levels and actions of plant hormones. The levels of auxin, gibberellins, cytokinins, ABA, ethylene and other phytohormones such as brassinosteroids, jasmonates and salicylates are affected by the action of melatonin through up- or downregulation of transcripts of some biosynthesis/catabolism enzymes, and also hormone-related regulatory factors; some interesting reviews about this can be consulted [26,32,36,37]. Photosynthesis, photorespiration and stomatal regulation, which are key pieces of the water and carbon economy in plants, are strongly regulated by phytomelatonin. Moreover, the metabolic pathways of carbohydrates, lipids and nitrogen and sulfur compounds are modulated by phytomelatonin, including the osmoregulatory response in stressful situations [38,39].

In this review, the most relevant works with Brassicaceae related to food and melatonin are analyzed. The role of melatonin as a beneficial agent in the postharvest preservation of broccoli is especially studied. Its beneficial effect on organoleptic aspects of this widely consumed vegetable is of great interest in the agri-food industry. In addition, an integrated model is presented showing in detail the role of melatonin on the biosynthesis of glucosinolates, its regulatory action on these compounds with anti-oncogenic activity and its high importance in our diets.

## 2. Brassicaceae Plants and Melatonin Studies

The family Brassicaceae, also known as Cruciferae, includes approximately 372 genera and 4060 species. It is distributed throughout the world, and its distribution pattern suggests that this plant family originated and was diversified from the eastern Mediterranean. Brassicaceae is a large family of plants that have a wide range of applications, including human consumption in vegetables, seed oils and condiments, livestock in fodder and others. *Arabidopsis thaliana*, which is well known in the research, belongs to this family. Among the vegetables that stand out for their high consumption are broccolis, cauliflowers, cabbages, collard greens, turnips and radishes; all of these derive from cultivars of *Brassica rapa* L. and *Brassica oleracea* L., which present a multitude of varieties and/or subspecies that provide an enormous culinary richness in the different regions where they are grown. *Brassica napus* L. *var oleracea* is cultivated worldwide for rapeseed oil. The three types of mustard: yellow (*Sinapsis alba* L.), black (*Brassica nigra* L.) and brown or Indian (*Brassica juncea* L.) are also widely cultivated and appreciated in gastronomy [40,41].

Table 1 shows the species of the Brassicaceae family that have been studied in melatonin treatments with food interest. In this case, only studies in seeds, seedlings and plants are shown in Table 1. Studies of *A. thaliana* and melatonin are not included in this review as we focus on Brassicacea of interest as foods.

Multiple studies of melatonin with A. thaliana can be consulted. In general, melatonin has a vegetative-growth-promoting effect, activating the biosynthesis of plant hormones, and balancing redox homeostasis, especially in situations of abiotic stress. A clear effect of Brassicaceae is the activation of anthocyanin biosynthesis, together with a foliar senescence-retarding effect. Therefore, melatonin treatments in plants, either foliar or root, result in plants of larger size, biomass and color and those with greater tolerance to stress—all very interesting aspects to be applied in the crops of the different Brassicaceae (see Table 1).

## 3. Postharvest Application of Melatonin in Brassicaceae

One of the most interesting aspects in studies on melatonin is its ability as a biomodulator of ripening and senescence. Many studies in this regard have been carried out, obtaining interesting results in the postharvest treatments of fruits such as apple, pear, tomato, cucumber, grape, plum, peach, apricot, cherry, strawberry, pomegranate, banana, etc., and in flowers such as carnation, anthurium, devil’s trumpet and peony. Some recent reviews on this topic, in which multiple data are analyzed and a scheme of action of melatonin on postharvest conservation is proposed, can be consulted [25,39,58,59,60].

At Brassicaceae, there are postharvest studies of melatonin application only on broccoli florets/heads (Table 2).

Melatonin treatment in broccoli florets shows interesting results for the postharvest conservation of this vegetable frequently consumed worldwide (Table 2). Visual and quality characteristics are clearly improved by treatments around 100 μM melatonin. Short treatments by immersion or spray induce a lower loss of chlorophylls and carotenoids, and therefore, an improvement in the chromatic spectrum of broccoli florets, as can be seen by a greater degree of hue angle (greener, bluer, less yellow). This anti-senescence effect was first described in 2009 by the authors in barley leaves [23], and has subsequently been verified in a multitude of plant species, both edible and wildtype, where an inhibitory effect of melatonin on activating senescence transcription factors was demonstrated [18,27,32,39,67,68]. In addition, broccoli florets preserve freshness better, with less hydric and texture loss. In general, the maintenance and cold transport of broccoli florets is essential for an acceptable shelf life. Temperatures around 4 °C are ideal for reaching a maximum duration of about 20–22 days. Melatonin treatments clearly enhance visual and quality characteristics, improving color, texture and shine, and are able to extend its shelf life between 5–8 days with maximum quality. Other parameters such as soluble solids, total acid content, astringency, bitterness and vitamin C content are improved with melatonin treatments (Table 2).

Regarding the compounds of nutraceutical interest in broccoli florets, melatonin treatments increase the contents of total phenols and flavonoids, and some specific ones such as rutin and quercetin (Table 2). Melatonin induces the expression of diverse transcripts of phenolic metabolism. In white and red cabbage seedlings, melatonin upregulates phenolic-related genes such as PAL, C4H, CHS, CHI, F3H, DFR and UFGT, among others [7]. Moreover, in fruits such as berries and kiwifruit, melatonin induces several flavonoid biosynthesis genes [28,29]. Melatonin has also been found to cause a general activation of the primary and secondary metabolism that leads to high levels of carbohydrates, lipids and amino acids, and a better energy level and redox status [38,39].

A characteristic of Brassicaceae, though not exclusive to it, is its relevant level in glucosinolates, which are a group of secondary metabolites. Evolutionarily, glucosinolates originated twice, so that they are found in two unrelated lines of plants: in the Brassicales order (mainly Brassicaceae, Capparaceae and Caricaceae families) and in the Putranjivaceae family [69]. These compounds contain N- and/or S- in their chemical structure, formulated as β-thioglucoside-*N*-hydroxysulfates. More than 120 glucosinolate compounds have been identified, and they can be classified as aliphatic glucosinolates (originating from methionine, valine, leucine or isoleucine), indole glucosinolates (originating from tryptophan) and aromatic glucosinolates (from phenylalanine or tyrosine) [70]. These secondary metabolites are relevant in the defense system of plants due their fungicide, bactericide, nematicide and allelopathic properties [71]. Moreover, Brassicaceae’s powerful odor and taste (“*mustard oil bomb*”) seems to repel herbivores, defending the plant from excessive consumption [72]. Likewise, the potent cancer chemoprotective and/or anti-oncogenic activity of glucosinolates and isothiocyanates (their hydrolysis products) has been described, promoting the cultivation and human consumption of glucosinolate-containing plant species as healthy foods [70,73,74].

In an experiment on broccoli florets without cooling, the contents of total aliphatic and indolic glucosinolates declined by more than 50% after 3 days of storage. However, the decrease was alleviated by melatonin treatment [63]. In broccoli florets stored at 20 °C [63] and also at 4 °C [65], melatonin induced total glucosinolate content, including the biosynthesis of several aliphatic glucosinolates such as GER and GRA, and decreased others such as GNA and PRO; moreover, indolic glucosinolates such as GBS, NGBS and 4MGBS were increased. Sulforaphane (SFR), an isothiocyanate product, was additionally increased (Figure 1). Melatonin upregulated the expression of several genes related to glucosinolate biosynthesis such as the transcription factors MYB28 and MYB34, and several transcripts of glucosinolate biosynthesis enzymes such as GS-*Elong*, UGT74B1, ST5b, FMOGS-OX1 and TGG1, and CYP83A1, CYP79F1 and CYP79B2, but AOP2 and ESP were downregulated by melatonin. The upregulation by melatonin of MYO/TGGs induced glucosinolate hydrolysis, reflecting the accumulated levels of sulforaphane in stored broccoli (Figure 1) [63,65]. In addition, in Chinese cabbage, a similar glucosinolate biosynthesis activation by melatonin was observed. In this case, an interesting study on the fungicide activity of glucosinolate-rich extracts against *Sclerotinia sclerotiorum* (stem rot disease) and its relationship with the melatonin-inducing capacity of glucosinolate biosynthesis was demonstrated [53]. An overall view is provided in the proposed model of Figure 1, but although we know that during the storage of broccoli glucosinolates are synthesized [75], and that melatonin is able to over-activate the biosynthesis of these compounds, it is important to point out that the conditions of temperature and light/dark determine the balance between glucosinolate biosynthesis and hydrolysis up to isothiocyanates, which considerably alter the flavors of the product by the time it reaches the consumer.

## 4. Conclusions

Although there are not many works on Brassicaceae plants, the existing ones confirm the bio-stimulating role of melatonin in plants under stress conditions [27,29,76]. Moreover, melatonin is also an activator of germination in old seeds or seeds exposed to contaminants (Table 1). The interconnection between plant primary and secondary metabolisms that melatonin establishes has been revealed in some studies [39]. In Brassicaceae such as *A. thaliana*, Chinese cabbage seedlings and broccoli florets, the promoting role in glucosinolate biosynthesis has been demonstrated. Melatonin treatments induce the accumulation of total aliphatic and indole glucosinolates, increasing specific glucosinolates, including GER, GRA, GBS and NGBS, among others (Figure 1). This glucosinolate biosynthesis induction by melatonin is possibly the result of an activation of the metabolism of S and N in these plants, but it is also due to the upregulation of specific transcription factors and enzymes to the glucosinolate pathway. The increased biosynthesis of glucosinolates also results in a higher content in their hydrolysis products—isothiocyanates. In addition to the interest in these compounds as anti-oncogenic agents, the use of melatonin is presented as a very important tool in fresh-cut broccoli, since various studies have shown that the application of melatonin in broccoli florets extends their shelf life and improves the postharvest quality of fresh-cut broccoli, which thereby maintain their green color and texture longer than untreated material, and also maintain higher levels of glucosinolates (Table 2, Figure 1).

Nevertheless, further investigations are needed: (i) to know better the effects of melatonin on primary metabolism, specifically on the photosynthetic, respiratory and carbohydrate transformation pathways (simple sugars, sucrose and starch, mainly); (ii) to form a better understanding of the regulatory mechanism of melatonin on glucosinolate metabolism (the aliphatic/indolic glucosinolate ratio and hydrolysis products); (iii) to know the possible effect of melatonin on mineral, dietary fiber and vitamin composition in plants and cabbages; (iv) postharvest, to better understand the effect of melatonin on the metabolism of ethylene and other plant hormones in broccoli; and (v) to study the effect of melatonin in other postharvest products of the Brassicaceae family such as cauliflower, Romanesco, kohlrabi, red cabbage and leafy vegetables.

## Figures and Tables

**Figure 1 molecules-27-01523-f001:**
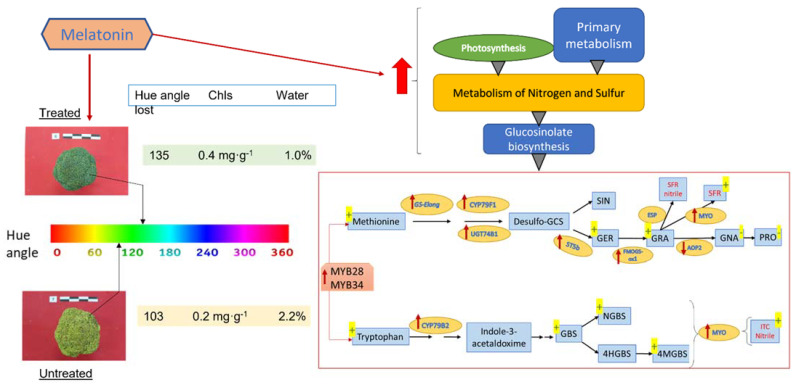
Representative model of melatonin’s effects on broccoli florets postharvest. Red arrows mean up-/downregulation, and yellow +/− means higher/lower levels with respect to untreated broccoli heads.

**Table 1 molecules-27-01523-t001:** Brassicaceae plants used in melatonin studies.

	Plant Species/Common Name	Melatonin Treatment	Response/Effect	References
* 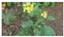 *	*B. juncea* L. Indian mustard seedlings	0.01–0.5 µM	↑ IAA, root growth	[42]
	*B. juncea var. gemmifera*Baby lateral buds	100 µM	↑ total phenols and glucosinolates, vit. C, carotenoids, ↓ weight loss, Chl loss	[43]
* 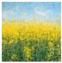 *	*B. napus var. oleracea*Rapeseed seedlings	50 µM	↑ growth, salt stress tolerance, IAA, ABA, BR and JA signaling factors, JA and BR levels	[44]
		0.01–100 µM	↑ salt tolerance, growth, redox balance	[45]
		500 µM	↑ drought tolerance, germination, Chl level, stoma size, redox balance	[46]
* 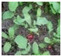 *	*Raphanus sativus var. radculus*Cherry radish seedlings	50–290 µM	↑ IAA, growth, heat stress tolerance, biomass, Chl levels, protein content, solid soluble content, redox balance	[47]
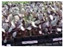	*B. oleracea var. rubrum*Red cabbage seedlings	1–100 µM	↑ germination, growth, Cu tolerance	[48]
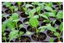	*B. oleracea var. album and rubrum*White and red cabbage seedlings	0.1–1 mM	↑ growth, anthocyanins, redox balance	[49]
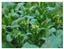	*B. rapa var. parachinensis*Chinese flowering cabbage (Choy Sum) seedlings	100 µM	↑ shelf life, energy level, ↓ ABA level, senescence factors, ABA biosynthesis genes, ABA-transcription factors, Chl-degrading genes, ROS, MDA, RBOH	[50,51,52]
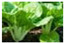	*B. rapa var. pekinensis* *Chinese cabbage*	50–100 µM	↑ sclerotinia rot tolerance, thiamine, ATP, glucosinolates, antioxidant enzymes	[53]
* 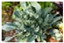 *	*B. oleracea var. italica*Broccoli seedlings	60 ppm	↑ growth, photosynthesis, biomass, Chl and carotenoid levels	[54]
		10 µM	↑ growth, Zn tolerance, glucosinolate biosynthesis genes, myrosinase, isothiocyanate, sulforaphane, ↓ EC, MDA	[55]
* 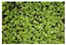 *	*Lepidum sativum* L. Gardencress seedlings	5–100 µM	↑ growth, Chl, carotenoid, anthocyanin and phenol levels	[56]
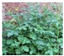	Mustard seeds	-	To obtain phytomelatonin	[57]

↑: Increased content or increased action; ↓: Decreased content or decreased action.

**Table 2 molecules-27-01523-t002:** Studies on melatonin application in broccoli (*B. oleracea var. italica*) heads/florets.

Plant Material/T^a^	Melatonin Treatment	Response/Effect	Ref.
Intact florets, 20 °C	100 µM immersed	↑ shelf life, Chls, ATP, ADP, SOD, CAT, POD ↓ yellow index, senescence, respiration rate, TCA, AMP, ROS	[61]
Intact florets, 20 °C	100 µM sprayed	↑ shelf life, Chls, flavonoids, carotenoids ↓ yellow index, senescence, bitterness, astringency, sulfur-volatiles, sulforaphane	[62]
Intact florets, 20 °C	1, 50, 500 µM immersed (5 min)	↑ shelf life, visual quality, Chls, carotenoids, vit. C, phenols, TAA, total glucosinolates, glucoraphanin, glucosinolate biosynthesis genes	[63]
Small cut florets, 4 °C	10, 100, 500 µM immersed (10 min)	↑ shelf life, Hue angle, Chls, FW, vit. C, TAA, phenols, flavonoids (rutin, quercetin, epicatechin), SOD, CAT ↓ yellow index, senescence, POD, MDA, ROS	[64]
Small cut florets, 4 °C	100 µM immersed (10 min)	↑ total glucosinolates, sulforaphane and glucoraphanin content, glucosinolate biosynthesis genes, myrosinase	[65]
Intact florets, 20 °C	100 µM immersed (30 min)	↑ shelf life, Chls, chloroplast integrity ↓ yellow index, Chl-degrading enzymes and genes	[66]

↑: Increased content or increased action; ↓: Decreased content or decreased action.

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
