# Peer review of "Melatonin in Brassicaceae: Role in Postharvest and Interesting Phytochemicals"

_molecules, 2022, doi:10.3390/molecules27051523_

Round 1
Reviewer 1 Report
The manuscript of Hernández-Ruiz et al. describes the influence of melatonin on a plant metabolism and shows its important role for a post-harvest conservation of vegetables. The review is interesting for fundamental understanding of mechanisms of the melatonin influence on plant and, additionally, for understanding of ways of its practical application. However, I have some minor comments:
- How much time is necessary for melatonin decomposition? How often plants or vegetables should be sprayed by melatonin in fields or during the post-harvest conservation?
- The melatonin can be consumed by animals and transmit into ground water. Can the spraying of melatonin be dangerous for environment? Is this problem considered in literature?
- Potentially, the melatonin can positively influence weed plants and increase its competitiveness in comparison with culture plants. Is this problem considered in literature?
- The melatonin can get into human organism when eating of vegetable which were sprayed by phytohormone. Is this problem considered in literature?
- Check the last names in literature (for example references 4, 17-26 and others), some signs are incorrect.
Author Response
Rev #1
The manuscript of Hernández-Ruiz et al. describes the influence of melatonin on a plant metabolism and shows its important role for a post-harvest conservation of vegetables. The review is interesting for fundamental understanding of mechanisms of the melatonin influence on plant and, additionally, for understanding of ways of its practical application. However, I have some minor comments:
- How much time is necessary for melatonin decomposition?
Response: Although the catabolism of melatonin in vitro is well known, we have no data on its decomposition rate in vivo in plants.
- How often plants or vegetables should be sprayed by melatonin in fields or during the post-harvest conservation?
Response: In general, field applications are made looking for plant growth objectives and resistance to stress; on the other hand, post-harvest treatments seek improvements in conservation and resistance to chilling.
- The melatonin can be consumed by animals and transmit into ground water. Can the spraying of melatonin be dangerous for environment? Is this problem considered in literature?
- Potentially, the melatonin can positively influence weed plants and increase its competitiveness in comparison with culture plants. Is this problem considered in literature?
- The melatonin can get into human organism when eating of vegetable which were sprayed by phytohormone. Is this problem considered in literature?
Response: There are problems to study of great relevance. Melatonin is classified as a non-harmful or dangerous compound, but no specific data in plants are available for now. In our paper (https://doi.org/10.1002/jsfa.11318) you will find our suggestions in this regard. Regarding the last point, we are conducting studies to evaluate the melatonin contents that can be ingested with post-harvest treatments in various vegetables.
- Check the last names in literature (for example references 4, 17-26 and others), some signs are incorrect.
Response: Sorry for the mistake due to errors in the text editor. Everything has been corrected.
Reviewer 2 Report
The article “Melatonin in Brassicaceae: role in postharvest and interesting
Phytochemicals” explains about the most relevant works with Brassicaceae related to food and melatonin. The article is interesting and well written. However for the comprehension of the readers, I would like to advise authors to add a small section about the “biogenesis of melatonin in plants”. For this the authors are advised to take benefit form the following article.
- 3390/ijms20051040
Moreover, in introduction section, the authors mentions about the efficacy of melatonin in human studies. It look un-justified. Please stay on your topic and only provide importance of melatonin in plants and plants based studies.
Table 1 and table 2 includes many abbreviations that need to be named fully when discussed first time. I advise authors to add a footer to the tables explaining all these. Similarly cross check all the manuscript for such details. I can not pin point each one as the manuscript donot include any line numbers.
I donot agree with the figure 1. As it is a review manuscript. The figure should not specify any specie such as broccoli. Authors should revise this figure and present a general overview explaining how melatonin performs its role in various mechanisms and regulates transcription factors. To have a general idea, see following paper please
- 3390/molecules26040862
Please do not refer to any citation or figure or table in conclusion section. Change this section into “conclusion and future prospects” section and provide the suggestions and future research directions regarding melatonin here.
Author Response
Rev.2
The article “Melatonin in Brassicaceae: role in postharvest and interesting phytochemicals” explains about the most relevant works with Brassicaceae related to food and melatonin. The article is interesting and well written. However, for the comprehension of the readers, I would like to advise authors to add a small section about the “biogenesis of melatonin in plants”. For this the authors are advised to take benefit from the following article.
- 3390/ijms20051040
Response: We appreciate the suggestion. However, we do not believe it is necessary to incorporate the elements of the melatonin biosynthesis pathways in plants, since our review deals with "the effects of exogenous melatonin in pre- and post-harvest time", without referring to endogenous levels of phytomelatonin. Undoubtedly, in many of our papers, readers can find out in detail the details of melatonin biosynthesis in plants.
Moreover, in introduction section, the authors mention about the efficacy of melatonin in human studies. It look un-justified. Please stay on your topic and only provide importance of melatonin in plants and plants based studies.
Response: From our point of view, we cannot ignore the origin of the studies on melatonin. In addition, in the case of post-harvest treatments, it is of interest to know the possible exposure to this molecule in human consumption, as other referees have expressed. We believe that this minimal appointment in the Introduction section helps to know the scope of application of melatonin.
Table 1 and table 2 includes many abbreviations that need to be named fully when discussed first time. I advise authors to add a footer to the tables explaining all these. Similarly cross check all the manuscript for such details. I can not pin point each one as the manuscript do not include any line numbers.
Response: A section of Abbreviations appears at the end of the manuscript.
I do not agree with the figure 1. As it is a review manuscript. The figure should not specify any specie such as broccoli. Authors should revise this figure and present a general overview explaining how melatonin performs its role in various mechanisms and regulates transcription factors. To have a general idea, see following paper please
- 3390/molecules26040862
Response: Our review focuses on the Brassicaceae family, and especially on the effects of melatonin on post-harvest products, of which broccoli is an excellent representative model, and the most studied with melatonin, as reflected in the title of the manuscript.
Please do not refer to any citation or figure or table in conclusion section. Change this section into “conclusion and future prospects” section and provide the suggestions and future research directions regarding melatonin here.
Response: We prefer to keep the Discussion section, usual in this journal.
Reviewer 3 Report
This is my feedback to the article: “ Melatonin in Brassicaceae: role in postharvest and interesting phytochemicals”, Josefa Hernández-Ruiz, Domingo Ruiz-Cano, Manuela Giraldo-Acosta, Antonio Cano and Marino B. Arnao
The review is well written, covers a wide time span, and includes the work of many famous scientists, including Ivana Machackova. The review is well illustrated with tables and pictures.
The article is entirely consistent with the content of the journal articles.
This area of research is very interesting and important, and I propose to accept the article after addition by the recently published articles :
https://www.ncbi.nlm.nih.gov/pmc/articles/PMC8832036/
https://doi.org/10.1093/plphys/kiab481
https://www.ncbi.nlm.nih.gov/pmc/articles/PMC8831713/
Author Response
Rev. 3
This is my feedback to the article: “Melatonin in Brassicaceae: role in postharvest and interesting phytochemicals”, Josefa Hernández-Ruiz, Domingo Ruiz-Cano, Manuela Giraldo-Acosta, Antonio Cano and Marino B. Arnao
The review is well written, covers a wide time span, and includes the work of many famous scientists, including Ivana Machackova. The review is well illustrated with tables and pictures.
The article is entirely consistent with the content of the journal articles.
This area of research is very interesting and important, and I propose to accept the article after addition by the recently published articles:
https://www.ncbi.nlm.nih.gov/pmc/articles/PMC8832036/
https://doi.org/10.1093/plphys/kiab481
https://www.ncbi.nlm.nih.gov/pmc/articles/PMC8831713/
Response: Thanks for your comments. We have incorporated the suggested paper related to the theme of the review in Table 1.